# Body Composition Assessment in Mexican Children and Adolescents. Part 2: Cross-Validation of Three Bio-Electrical Impedance Methods against Dual X-ray Absorptiometry for Total-Body and Regional Body Composition

**DOI:** 10.3390/nu14050965

**Published:** 2022-02-25

**Authors:** Desiree Lopez-Gonzalez, Jonathan C. K. Wells, Patricia Clark

**Affiliations:** 1Clinical Epidemiology Research Unit, Hospital Infantil de México Federico Gomez, Mexico City 06720, Mexico; dradesireelopez@gmail.com; 2Faculty of Medicine, Universidad Nacional Autonoma de México, Mexico City 06720, Mexico; 3Childhood Nutrition Research Centre Population, Policy and Practice, UCL Great Ormond Street Institute of Child Health, London WC1N EH, UK; jonathan.wells@ucl.ac.uk

**Keywords:** body composition, bioelectric impedance analysis (BIA), children, adolescents, validation, equation

## Abstract

The aim of our study was to validate three different bioelectrical impedance analysis (BIA) methods for estimating body composition (BC). First, we generated BIA prediction equations based on the 4-C model as the reference method for fat mass (FM) and fat-free mass (FFM), and on dual X-ray absorptiometry (DXA) estimations of appendicular lean mass (ALM) and truncal fat mass (tFM). Then, we performed cross-validation in an independent BMI-, sex-, and Tanner-stratified sample of 450 children/adolescents. The three BIA methods showed good correlation and concordance with DXA BC estimations. However, agreement analyses showed significant biases, with increasing subestimations of FM and tFM, and overestimations of ALM, by all three BIA methods. In conclusion, the three BIA methods analysed in this study, provide valid estimations of BC for total body and body segments, in children and adolescents who are of a healthy weight, overweight, or obese. It should be noted that this validation cannot be extrapolated to other BIA methods.

## 1. Introduction

Body composition (BC) assessment refers to the estimation of the relative distribution of the different components that constitute a human body (e.g., fat, lean mass, bone, and water mass) [1]. Normal variation of BC occurs in association with gender, age, pubertal status, and ethnicity; therefore, ideally, population-specific reference values should be available. BC assessment is increasingly considered in the clinical care of children and adolescents, mainly because several health conditions are related to abnormalities in BC (e.g., obesity, malnutrition, cancer, growth problems, chronic exposure to steroids, etc.) [2,3].

There are several methods to assess BC, ranging from simple to complex approaches, each of which has advantages, limitations, and some degree of measurement or estimation error [4]. The gold standard of BC measurement is cadaver analysis; hence, all in vivo techniques may be considered approximations of this criterion [3,5]. The four-component (4-C) model represents one of the most accurate in vivo approaches of BC estimation, though it is expensive and usually available only in the context of specialized research [4,6]. Ideally, BC estimation methods should assess the distribution of tissues by body regions or segments (i.e., trunk and limbs). Thus, multi-frequency bioelectrical impedance analysis (BIA) is emerging as an ideal method for assessing body composition at paediatric ages [7], both in the context of research and routine clinical use, due to its practicality, low cost, the availability of results by body segments, little need for training to perform the measurement, and safety.

Currently, several BIA devices are available for estimating BC in children and adolescents [8,9,10]. However, not all devices have been clinically validated and reference values are scarce.

The aim of our study was to generate and validate equations for the estimation of BC (i.e., fat mass (FM), fat-free mass (FFM), truncal fat mass (tFM), and appendicular lean mass (ALM)) through BIA in children and adolescents, using dual-energy X-ray absorptiometry (DXA) as a reference method.

## 2. Materials and Methods

### 2.1. BIA Equation Generation

BC data for the whole body was estimated using the 4C model, regional BC was estimated using DXA, TBW was estimated using D2O, and raw BIA data from the sample of 299 children and adolescents that participated in part 1 of this study, were used to generate BIA equations to estimate FM, FFM, ALM, and tFM. Measurement requirements, procedures, and characteristics of the study subjects have been described previously (Part 1 Nutrients 1601338).

### 2.2. BIA Measurements

Three different devices were used to acquire raw data on resistance (R), reactance (Xc), and phase angle (PA) at: (1) standing position BIA handrail (SECA mBCA 514 seca deutschland seca gmbh & co. kg. Hamburg, Germany), (2) standing position BIA handle (SECA modified mBCA 514 for the paediatric population), and (3) supine position BIA (SECA 525 seca deutschland seca gmbh & co. kg. Hamburg, Germany).

#### 2.2.1. Standing-Position MF-BIA Handrail and Paediatric Model

These models consist of a platform with an integrated scale and a handrail system. Each side of the handrail carries six electrodes, out of which two are chosen, depending on the participant’s height. According to the manufacturer’s instructions, arms should be held straight [11]. In the paediatric prototype, subjects hold the electrodes with their arms at an angle of 30 degrees from the body. Subjects ≥ 8-h fasting, in light clothing, without metallic accessories, and barefoot, were asked to maintain a standing position for at least 10 min before the measurement.

#### 2.2.2. Supine Position BIA

This model is designed for measurements in the supine position and can be operated using either 4 adhesive electrodes on the right side of the body (4e) or 8 electrodes (8e) on both sides of the body, while the subject is lying on a non-conductive surface. Both approaches were applied here. Adhesive electrodes were placed at defined anatomical sites on the dorsal surface of the hand, wrist, ankle, and foot, according to the manufacturer’s instructions, as follows: the proximal edge of the first electrode was attached by reference to an imaginary line at the styloid process of the ulna, whereas the distal edge of the finger electrode was attached on an imaginary line from the middle of the metacarpophalangeal joints of the index and middle fingers; however, if the distance between the electrodes would have been <5 cm, the distal electrode was placed on the palm. The distal edge of the toe electrode was placed by reference to an imaginary line through the middle of the metatarsophalangeal joints of the second and third toes. The proximal edge of the ankle joint electrode was attached along a line through the highest points of the outer and inner ankle bones (see Appendix A). Subjects ≥ 8-h fasting, in light clothing, without metallic accessories, and barefoot were asked to be at supine position for at least 10 min before 8e and 4e measurements took place.

Impedance was measured with a current of 100 μA, at frequencies between 1 and 1000 kHz, and the duration of each measurement was 75 s. Measurements in the supine position were taken after lying down for 10 min.

Resistance (R) and reactance (Xc) values, obtained at a frequency of 50 kHz, were used to generate the prediction equations. The impedance index was calculated as height2/R. Two indices, IndexR50 trunk/extremities and IndexXc50 trunk/extremities were derived from the segmental R and Xc values (the means of the left and right body side measurements), representing the relative contribution of the trunk and extremities to total body conductivity and correcting for differences in body shapes.

### 2.3. BIA Equations Validation

Cross-validation was performed in an independent stratified random sample of 450 children and adolescents, referred to as the “validation sample”. Stratification considered sex (male, female), three BMI categories (healthy weight, overweight, and obese), and sexual maturation according to Tanner stages (prepuberal = Tanner 1, puberal = Tanner 2 and 3, and late puberal = Tanner 4 and 5). As a result, each age/sex/maturation group included 25 subjects. This validation sample consisted of subjects who participated in the “Reference values for body composition of Mexican children and adolescents” and the “Reference values for bone mineral density in healthy Mexican children and adolescents“ study [12]. These 450 subjects were assessed for BC by means of DXA and the 3 BIA methods.

### 2.4. Statistical Analysis

Descriptive statistics were used to report demographics and the estimates for each method, and are expressed as means and standard deviations for continuous variables, and as percentages for categorical variables.

#### 2.4.1. Generation of BIA BC Prediction Equations

With the reference sample data (*n* = 299), prediction equations were generated by stepwise regression. Stepwise linear regression was performed using height2/(R50 weight), Xc50, IndexR50 trunk/extremities, R50/weight, height, weight, age, and gender as covariates. Dependent variables were chosen as values relative to body weight in order to obtain an evenly distributed relative error and, as a result, to ensure smaller absolute errors in young children. For the lean soft tissue (LST) of arms and legs, predictors had to be similar on the right and left sides.

For all BIA methods, resistance R and reactance Xc were measured at 50 kHz. The indices equations have been previously described [11]:$$Index\;R_{50\;trunk/extremities}=\frac{4\;R_{50\;trunk}}{R_{50\;right\;arm}+R_{50\;left\;arm}+R_{50\;right\;leg}+R_{50\;left\;leg}}$$
$$Index\;Xc_{50\;trunk/extremities}=\frac{4\;Xc_{50\;trunk}}{Xc_{50\;right\;arm}+Xc_{50\;left\;arm}+Xc_{50\;right\;leg}+Xc_{50\;left\;leg}}$$

#### 2.4.2. Cross-Validation

Estimations of FM, FFM, ALM, and tFM from the validation sample and by the three BIA methods were contrasted with those of DXA as the reference standard by means of Pearson correlation, linear regression, and Lin’s concordance coefficients.

The Bland-Altman method [13] was used to assess agreement between each BIA method with DXA as the reference standard. The differences of BC values estimated by each BIA device, minus the values estimated using DXA (y-axis), were plotted against the average of said two measurements (x-axis). Differences and limits of agreement were calculated. A linear regression analysis with differences as the dependent variable and the average of measurements as the independent variable was carried out for each BIA method in order to detect if there was a proportional difference [13].

Statistical analyses were performed using SPSS for Windows version 21.0 (SPSS Inc., Chicago, IL, USA), and Prism 8 for Windows (GraphPad Software, Inc., San Diego, CA, USA). Prediction equations were carried out using R software, version 3.0.1 (The R Foundation for Statistical Computing, Vienna, Austria). Statistical significance was set at *p* < 0.05 for all analyses.

## 3. Results

Data of 299 children and adolescents comprised the reference sample to generate the BIA BC prediction equations. The 4C model measurements, demographics, and general characteristics of these subjects are presented in Table 1.

Regression analyses of the reference sample data identified IndexR50 trunk/extremities as the strongest predictor, explaining 93% to 99% of the variance in TBW, FFM, tFM, and appendicular LST. As shown in Supplementary Appendix A, R² and standard errors of the estimates (SEE) are shown for each step of the regression, and significance is shown for the complete regression. For a comparison of accuracy with the existing formulas, R² and SEE are also shown for the reference measurement absolute value (not relative to body weight) (Supplementary Appendix A).

Demographics and general characteristics of the validation sample (*n* = 450) are presented in Table 2.

### Cross-Validation

The correlation of BC estimations using the three BIA methods and DXA were strong for all estimates (*r* ≥ 0.989) (Figure 1), though Lin’s concordance coefficients ranged from poor (ρc = 0.75) to almost perfect (ρc = 0.994) depending on the estimate and method (Table 2, Table 3, Table 4 and Table 5). Finally, Bland-Altman analyses, informed of the differential behaviours according to the different estimations and methods, were conducted (Table 3, Table 4, Table 5 and Table 6 and Figure 2, Figure 3, Figure 4 and Figure 5).

The correlation of FM estimations using the three BIA methods and DXA were strong from *r* = 0.97 to *r* = 0.99. The Lin’s concordance coefficients ranged from poor (ρc = 0.84) to substantial (ρc = 0.98), depending on the BMI category and method (Table 3)

FM means estimations were significantly lower for all three BIA methods compared to the DXA estimates for all BMI categories. In the BA analysis, differences ranging from 2.8 ± 1.1 (95% CI 2.7 to 2.9) for BIA supine 8e to 3.6 ± 1.5 (95% CI 3.4 to 3.7) for BIA handle, with significant bias (*p* < 0.001), resulting in greater underestimations with higher FM values (Figure 2).

When FFM was estimated, a high correlation was found in all BIA methods with respect to the DXA estimates (*r* ≥ 0.99); Lin’s concordance coefficients ranged from poor (ρc = 0.75) to substantial (ρc = 0.97), depending on the method (Table 4), and concordance for BIA handle and supine 8e was substantial in all three groups (ρc ≥ 0.95); however, it was low for BIA handrail, especially in the group with normal weight, and for supine 4e, in the OB group, as observed in Table 4.

In the BA analysis, the differences between the FFM BIA estimations vs. the DXA estimates, ranged from 2.8 ± 1.0 (95% CI 0.8 to 4.9), for BIA handle in the NW group, to 4.4 ± 1.8 (95% CI 0.8 to 7.97) for BIA Handrail in the OB group, with significant bias (*p* < 0.001), which means that a higher value of FFM, results in more overestimations, Figure 3.

When BC was estimated by body region, a very high correlation was found in all BIA methods with respect to DXA (*r* ≥ 0.99) for ALM estimations; the Lin’s concordance coefficients were moderate (ρc = 0.87) to almost perfect (ρc = 0.99), with substantial values for BIA handle and supine 8e; however, by BMI categories, the results were r for BIA handrail (ρc = 0.77 in NW to ρc = 0.88 in OB), as observed in Table 5.

Comparing the ALM estimations between the different BIA methods and DXA, using BA analysis, the bias was 0.02 ± 0.59 (95% CI 0.17 to 0.63) for BIA supine 8e to 0.61 ± 0.65 (95% CI −0.66 to 1.87) for BIA handle, in the total sample (Figure 4). The bias in ALM estimation, showed a significant negative correlation between bias and the mean ALM estimated using the BIA methods, which indicates that higher ALM values, result in more overestimations (Table 5 and Figure 4).

When tFM was estimated correlation, concordance, and agreement between DXA and the BIA methods were strong (*r* ≥ 98), concordance (ρc ≥ 0.94), and bias of −0.23 ± 0.97 kg) for the entire sample. Concordance was also high for the tFM estimation with BIA handle and for both 8e and 4e BIA supine, with values ≥0.96 in all BMI categories (Table 5). However, it was lower for BIA handrail (ρc = 0.84). Only in respect of BIA handle and BIA supine 4e, was the bias equation not significant. Performance, by BMI category, is shown in Table 6.

An analysis of the clinical differences in the subgroup of participants measured with each BIA device was performed, from the validation sample. The main differences were found in the subgroup of subjects measured with BIA handrail for both anthropometric and body composition measurements because the device criteria required that subjects be of a height of >140 cm. Clinical characteristics and the body composition of the subjects measured with each BIA method are shown in Appendix A.

## 4. Discussion

This study evaluated three different BIA methods (handle, handrail, supine-position with 8 electrodes and 4 electrodes) for estimating the body composition of Mexican children and adolescents. First, we generated prediction equations based on the 4C model as the reference method for FM and FFM, and on DXA estimations for appendicular lean mass (ALM) and truncal fat mass (tFM). Second, we performed cross-validation in an independent sample of Mexican children and adolescents with different BMI categories (healthy weight, overweight, and obese), stratified by sex and different stages of sexual maturation.

While all BIA methods showed good correlation and concordance to DXA BC estimations, we found that BIA handrail had the lowest concordance. This might be because only older and taller children were measured on this device. The exclusion of young children reduces the variability of the respective body component. This could, in part, explain the lower r and R².

Previous studies considering the validity of different BIA methods in the paediatric population have produced inconsistent findings. Some of these studies report strong correlations with DXA, whereas others, present discordant results. The inconsistent findings may be related to the fact that different methods and populations have been evaluated, devices vary in terms of the measuring methodology, including the way the current passes through the body (foot to foot, hand to foot, or hand to hand), the number of electrodes employed (4 vs. 8), subject position (standing or supine), or the type of frequency used (single-frequency vs. multi-frequency), as well as the equations that are employed [14,15].

The validity of a method is closely related to the quality of the equations used in the estimation of BC; ideally, these should be generated from data of the gold standard. In addition, in relation to the paediatric population, adaptations of equations developed for adults should be avoided [3,16] because the results will vary according to the body surface area and the level of hydration [11].

In this study, the generation of the equations was based on the 4-compartment body composition model, using ADP, D2O, and DXA, and with cross-validation with DXA estimations.

The results show strong correlations in BC estimates between the BIA methods and DXA (*r* > 0.95); however, measurement bias was significant in the whole-body estimates. BIA overestimated FFM, but underestimated FM and FM% relative to DXA, which is similar to the results of other BIA devices that have been evaluated by reference to DXA [8,10,17]. Fat mass underestimation with BIA compared to that reported using DXA, maintained significant correlations between the bias and measurements average, which suggests that the higher an individual’s fat mass, the greater the underestimation of the BIA device [14,15].

Ideally, the evaluation of a subject’s BC should also include an analysis of tissue distribution; for example, quantifying central adiposity and muscle tissue in the extremities. Such estimation is of clinical utility, for instance, in the evaluation of a subject’s growth and development, when monitoring the effects of exercise or the impact of a treatment, or the effects of an illness or trauma. Multi-frequency bioelectrical impedance has been reported to be a useful method for making these estimates, although they have shown variable accuracy in children in previously published studies [16]. In our study, we found a high correlation and concordance for ALM estimation between the different BIA methods evaluated and DXA, for all BMI categories, with a higher agreement and less bias between BIA and DXA, compared to other publications [18].

It is important to note the consistency found in the different methods for full-body composition estimations, as well as by region, among different BMI categories. The estimation of BC by BIA can be used to monitor the effects of treatments in patients of normal body weight, or who are obese or overweight because it is a safe method, that is comfortable for the patient, and which has different measurement options (standing and supine, total or half body). BIA is also useful for monitoring individual children, over time.

For FM estimation, it will be necessary to perform a further validation taking the 4C model as the reference because, as we found in this study, BIA underestimates FM compared to the estimates produced by DXA; however, the results from the first part of this study (Part 1 Nutrients 1601338), show that DXA overestimates FM content when compared to the 4C model [19].

A possible explanation for the FM estimation discrepancy between BIA and DXA may be that the BIA equations were generated based on the 4C model. In the adult Hispanic population, the bias when estimating FFM using the 4C model was 0.4 ± 1.8 kg; in this study, the bias in the estimation of FFM using BIA in comparison with DXA, was higher (2.8 kg), although we should consider that the reference method is different in both studies (4C in adults and DXA in our study), as well as the age of the subjects (children in our study).

The results of this study indicate that BIA is a useful method for measuring BC and it has an equal ranking with DXA in terms of consistency. However, BIA should not be regarded as an interchangeable method with DXA, due to the bias between methods. Specifically, for patient monitoring, continuing with the initial assessment method is suggested. Since the ISCD indication for DXA in children is to perform the assessment with a frequency of less than every 6 months (for BMD, for BC there is still no agreed best practice in paediatrics), BIA can be a safe and reliable alternative for patient monitoring, provided possible biases are considered. BIA has been shown to be especially useful for the evaluation of body segments, such as appendicular lean mass and trunk fat mass.

### Limitations of This Study

The main limitation of this study was the use of DXA as the reference method because total BC estimation with DXA does not consider subjects’ hydration status, which is especially important in paediatrics due to the great variability across different age groups and depending upon nutritional status.

The validation of the BIA method results are only applicable to the devices included in this study and cannot be extrapolated to other BIA devices.

The generation of equations and their cross-validation were only for Mexican subjects; validation in other populations is still pending.

The strengths of this study are that it included a representative sample of children and adolescents from Mexico City, stratified by pubertal stage and considering different BMI categories. It simultaneously evaluated the performance of different BIA devices in two positions. Cross-validation was performed for estimations of total body BC, as well as for different body regions, such as trunk fat mass and appendicular lean mass.

## 5. Conclusions

Multi-frequency BIA in both the supine and standing positions can be a safe and reliable alternative for BC measurement in children and adolescents. It provides valid estimates of BC in children and adolescents who are of healthy weight, overweight, or obese; however, it is not interchangeable with estimations made with DXA for patient monitoring. Its main utility is in the evaluation of regional body composition, such as appendicular lean mass and trunk fat mass.

## Figures and Tables

**Figure 1 nutrients-14-00965-f001:**
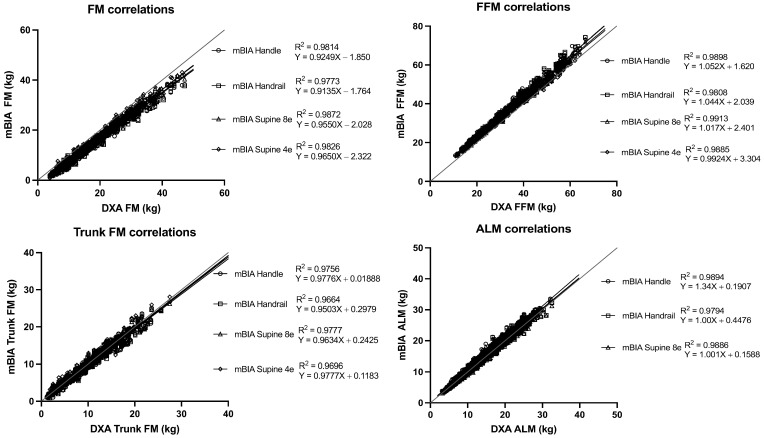
Linear regression analysis for BC measurements in the validation sample: total body FM and FFM; regional: ALM and tFM between the DXA and BIA methods (handle, handrail, and supine 8e and supine 4e). Abbreviations: DXA, dual X-ray absorptiometry; FM, fat mass; FFM, fat-free mass; ALM, appendicular lean mass; mBIA, multifrequency bioelectrical impedance analysis.

**Figure 2 nutrients-14-00965-f002:**
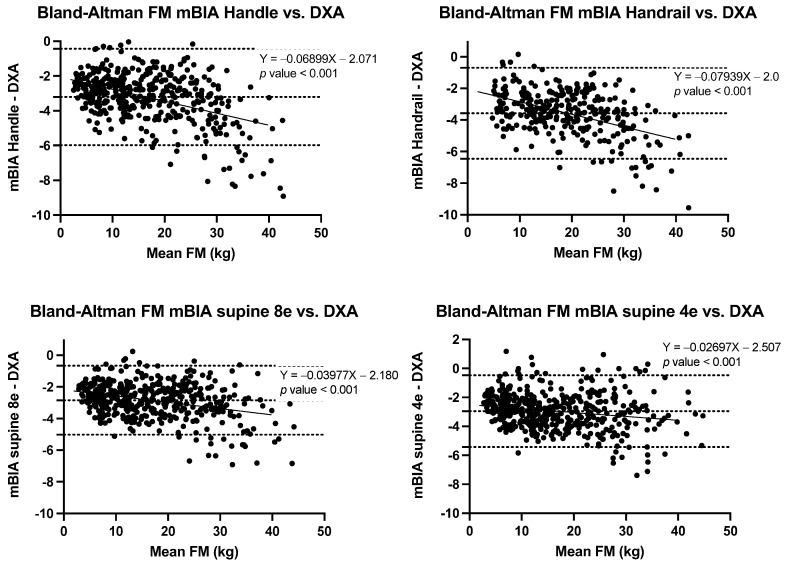
Bland-Altman plots with limits of agreement in FM estimation between BIA methods and DXA. Abbreviations: DXA, dual X-ray absorptiometry; FM, fat mass; mBIA, multifrequency bioelectrical impedance analysis.

**Figure 3 nutrients-14-00965-f003:**
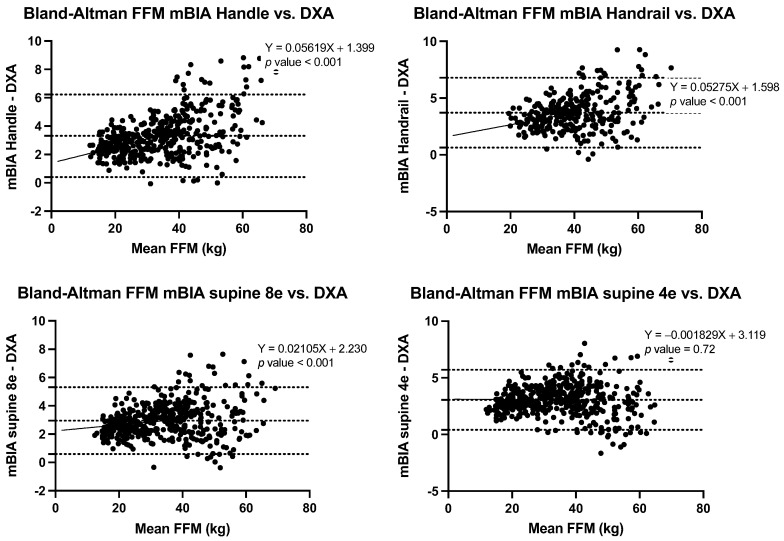
Bland-Altman plots of FFM estimation between BIA methods and DXA.

**Figure 4 nutrients-14-00965-f004:**
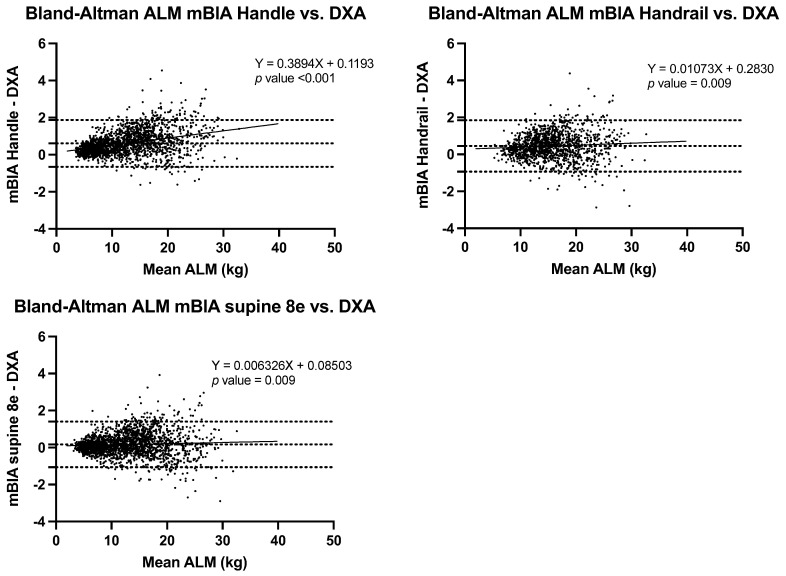
Bland-Altman plots of ALM estimation between BIA methods and DXA.

**Figure 5 nutrients-14-00965-f005:**
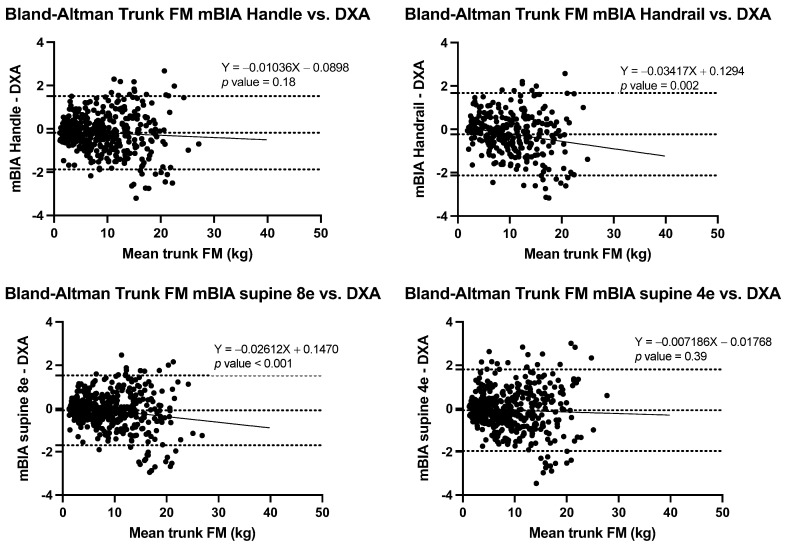
Bland-Altman plots of trunk fat mass estimation between BIA methods and DXA.

**Table 1 nutrients-14-00965-t001:** Demographics and general characteristics of the reference sample used to generate BIA BC prediction equations.

	Female (*n* = 158)		Male (*n* = 141)		Total (*n* = 299)	
	Mean ± SD	Range	Mean ± SD	Range	Mean ± SD	Range
Age (years)	11.9 ± 3.8	(4.8–18.0)	12.2 ± 3.7	(5.0–18.6)	12.0 ± 3.7	(4.8–18.6)
Weight (kg)	41.0 ± 16.3	(13.2–88.5)	43.3 ± 16.2	(17.1–77.3)	42.1 ± 16.3	(13.2–88.5)
Weight Z-score	−0.22 ± 0.99	(−2.45–2.19)	0.13 ± 1.17	(−1.68–4.49)	−0.06 ± 1.09	(−2.45–4.49)
Height (cm)	142.5 ± 18.2	(97.6–170.4)	148.7 ± 20.8	(105.9–186.0)	145.5 ± 19.7	(97.6–186.0)
Height Z-score	−0.48 ± 0.93	(−3.02–1.53)	−0.22 ± 0.90	(−2.96–2.46)	−0.36 ± 0.93	(−3.02–2.46)
BMI (kg/m^2^)	19.2 ± 4.1	(12.7–35.2)	18.7 ± 3.2	(13.4–27.8)	19.0 ± 3.7	(12.7–35.2)
BMI Z-score	0.24 ± 1.09	(−2.88–2.87)	0.14 ± 1.19	(−2.98–5.15)	0.19 ± 1.14	(−2.98–5.15)
Waist circumference (cm)	65.1 ± 10.1	(46.7–92.3)	66.7 ± 10.2	(47.5–91.9)	65.9 ± 10.1	(46.7–92.3)
Tanner puberal stage						
1	51 (32%)		62 (44%)		113 (37%)	
2 & 3	48 (30%)		44 (31%)		92 (31%)	
4 & 5	59 (37%)		35 (25%)		94 (22%)	
BMI category						
Low weight	2 (1%)		3 (2%)		5 (2%)	
Healthy weight	117 (74%)		109 (77%)		226 (76%)	
Overweight	30 (19%)		21 (15%)		51 (17%)	
Obesity	9 (6%)		8 (6%)		17 (6%)	
4C FM (kg)	10.8 ± 7.7	(0.4–38.6)	8.1 ± 5.6	(0.4–26.0)	9.5 ± 6.9	(0.4–38.6)
4C FFM (kg)	30.1 ± 9.8	(12.4–53.3)	35.3 ± 13.2	(14.9–65.9)	32.5 ± 11.8	(12.4–65.9)
ADP volume (L)	39.4 ± 16.1	(12.0–87.9)	41.0 ± 15.4	(15.7–75.1)	40.1 ± 15.8	(12.0–87.9)
D2O TBW (L)	21.7 ± 7.2	(8.0–37.5)	25.5 ± 9.4	(10.3–48.6)	23.5 ± 8.5	(8.0–48.6)
DXA measurements						
BMC (kg)	1.5 ± 0.6	(0.4–2.7)	1.6 ± 0.7	(0.6–3.4)	1.6 ± 0.6	(0.4–3.4)
FM (kg)	13.7 ± 7.4	(3.2–41.0)	10.8 ± 5.6	(3.3–29.7)	12.3 ± 6.8	(3.2–41.0)
FFM (kg)	27.2 ± 9.4	(9.6–49.4)	32.5 ± 12.9	(12.4–60.6)	29.7 ± 11.5	(9.6–60.6)
Lean mass (kg)	25.7 ± 8.9	(9.2–46.9)	30.9 ± 12.2	(11.7–57.3)	28.1 ± 10.9	(9.2–57.3)
ALM (kg)	11.3 ± 4.5	(3.1–22.3)	14.1 ± 6.3	(4.3–27.5)	12.6 ± 5.6	(3.1–27.5)
BIA Handle	*n* = 36		*n* = 41		*n* = 77	
PA Handle	4.3 ± 0.5	(3.5–5.6)	4.5 ± 0.6	(3.3–5.9)	4.4 ± 0.5	(3.3–5.9)
R50T Handle (Ω)	30.5 ± 3.3	(23.4–36.3)	28.9 ± 3.5	(21.5–36.1)	29.7 ± 3.5	(21.5–36.3)
Xc50T Handle (Ω)	−3.3 ± 0.3	(−4.1–−2.8)	−3.5 ± 0.4	(−4.5–−2.8)	−3.4 ± 0.4	(−4.5–−2.8)
BIA Handrail	*n* = 129		*n* = 117		*n* = 246	
PA Handrail	4.8 ± 0.5	(3.7–6.4)	5.0 ± 0.7	(3.7–6.8)	4.9 ± 0.6	(3.7–6.8)
R50T Handrail (Ω)	28.1 ± 3.9	(21.5–43.3)	26.5 ± 4.1	(20.0–37.1)	27.3 ± 4.1	(20.0–43.3)
Xc50T Handrail (Ω)	−2.9 ± 0.4	(−3.8–−2.0)	−3.1 ± 0.4	(−4.5–−1.9)	−3.0 ± 0.4	(−4.5–−1.9)
BIA supine 8 electrodes	*n* = 150		*n* = 140		*n* = 290	
PA mBIA supine 8E	5.5 ± 0.6	(4.2–7.4)	5.7 ± 0.9	(3.8–8.3)	5.6 ± 0.8	(3.8–8.3)
R50T mBIA supine 8E (Ω)	31.3 ± 3.7	(24.1–43.1)	30.3 ± 4.1	(22.7–41.7)	30.8 ± 3.9	(22.7–43.1)
Xc50T mBIA supine 8E (Ω)	−3.8 ± 0.4	(−4.9–−3.0)	−4.0 ± 0.4	(−4.9–−2.8)	−3.9 ± 0.4	(−4.9–−2.8)
BIA supine 4 electrodes RS	*n* = 158		*n* = 141		*n* = 299	
PA mBIA supine 4E RS	5.5 ± 0.7	(4.0–7.5)	5.7 ± 0.9	(3.8–8.5)	5.6 ± 0.8	(3.8–8.5)
R50right side mBIA supine 4E (Ω)	768.2 ± 98.9	(589.4–1032.2)	682.3 ± 100.7	(488.2–903.8)	727.7 ± 108.5	(488.2–1032.2)
Xc50right side mBIA supine 4E (Ω)	−73.5 ± 8.9	(−99.1–−56.5)	−67.6 ± 7.4	(−87.1–−44.1)	−70.7 ± 8.7	(−99.1–−44.1)

Data shown as mean and standard deviation (SD). Abbreviations: BMI, body mass index; DXA, dual X-ray absorptiometry; BMC, bone mineral content; FM, fat mass; FFM, fat-free mass; ALM, appendicular lean mass; mBIA, multifrequency bioelectrical impedance analysis; TBW, total body water; R, resistance; Xc, reactance; and PA phase angle.

**Table 2 nutrients-14-00965-t002:** Demographics and general characteristics of the validation sample.

	Female (*n* = 225)		Male (*n* = 225)		Total (*n* = 450)	
	Mean ± SD	Range	Mean ± SD	Range	Mean ± SD	Range
Age (years)	11.6 ± 3.7	8.6–14.3	12.4 ± 3.6	9.6–15.2	12.0 ± 3.7	4.7–19.0
Weight (kg)	47.7 ± 18.9	31.8–61.9	53.9 ± 21.2	36.7–67.3	50.8 ± 20.3	14.5–108.5
Weight Z-score	0.69 ± 1.25	−3.19–4.09	0.76 ± 1.17	−2.17–3.54	0.73 ± 1.21	−3.19–4.09
Height (cm)	142.8 ± 16.8	128.0–155.9	151.6 ± 18.9	135.6–166.7	147.2 ± 18.4	101.8–184.8
Height Z-score	−0.3 ± 1.1	−0.95–0.5	−0.1 ± 1	−0.79–0.64	−0.16 ± 1.07	−3.01–4.09
BMI (kg/m^2^)	22.4 ± 5.1	18.1–26.1	22.5 ± 5.1	18.7–26.0	22.4 ± 5.1	12.9–37.1
BMI Z-score	0.98 ± 1.06	−2.45–2.9	0.92 ± 1.14	−2.07–3.28	0.95 ± 1.10	−2.45–3.28
Waist circumference (cm)	72.7 ± 12.6	62.4–82.7	76.5 ± 14.4	66.6–87.0	74.6 ± 13.7	45.0–117.0
Tanner puberal stage						
1	75 (33%)		75 (33%)		150 (33%)	
2 & 3	75 (33%)		75 (33%)		150 (33%)	
4 & 5	75 (33%)		75 (33%)		150 (33%)	
BMI category						
Healthy weight	75 (33%)		75 (33%)		150 (33%)	
Overweight	75 (33%)		75 (33%)		150 (33%)	
Obesity	75 (33%)		75 (33%)		150 (33%)	
DXA measurements						
BMC (kg)	1.6 ± 0.6	1.0–2.0	1.9 ± 0.7	1.2–2.4	1.7 ± 0.7	0.6–3.4
FM (kg)	18.8 ± 9.5	10.8–24.8	17.3 ± 9.4	9.4–24.0	18.0 ± 9.5	3.9–47.2
FFM (kg)	28.7 ± 10	19.5–36.2	36.4 ± 13.7	23.6–47.6	32.5 ± 12.6	10.8–66.6
Lean mass (kg)	27.1 ± 9.4	18.5–34.3	34.6 ± 13	22.5–45.0	30.8 ± 11.9	10.2–63.3
ALM (kg)	11.9 ± 4.8	7.7−15.4	15.8 ± 6.7	9.7–21.0	13.9 ± 6.1	3.3–32.5
tFM (kg)	9.3 ± 5.3	1.2–25.6	8.7 ± 5.7	1.2–27.5	9.0 ± 5.5	1.2–27.5
BIA Handle	*n* = 210		*n* = 215		*n* = 425	
FM mBIA Handle (kg)	15.6 ± 9.1	7.5–22.3	14.1 ± 8.7	6.4–20.5	14.8 ± 8.9	1.1–40.3
FFM mBIA Handle (kg)	31.8 ± 10.8	22.3–39.4	39.5 ± 14.3	26.7–50.6	35.7 ± 13.2	13.3–74.4
TBW mBIA Handle (L)	23.3 ± 7.3	16.8–28.8	28.7 ± 9.6	20.1–36.7	26.0 ± 8.9	9.5–49.8
ASMM mBIA Handle (kg)	12.4 ± 5.1	7.6–15.8	16.5 ± 6.8	10.0–21.4	14.1 ± 6.2	3.5–30.9
PA mBIA Handle	4.7 ± 0.5	4.4–5.0	5.1 ± 0.6	4.6–5.5	4.9 ± 0.6	3.7–6.9
R50T mBIA Handle (Ω)	27.4 ± 3.9	18.9–41.6	25 ± 3.9	16.7–34.7	25.4 ± 3.5	16.7–41.6
Xc50T mBIA Handle (Ω)	−3.4 ± 0.5	−5.4–−1.9	−3.5 ± 0.4	−4.6–−2.6	−3.5 ± 0.5	−5.4–−1.9
BIA Handrail	*n* = 139		*n*= 158		*n*= 297	
FM mBIA Handrail (kg)	18.8 ± 8.2	11.8–23.9	16.2 ± 8.2	9.8–22.4	17.4 ± 8.3	2.8–39.9
FFM mBIA Handrail (kg)	37.9 ± 8.4	31.9–43.0	45.4 ± 11.6	35.8–53.8	41.9 ± 10.9	21.2–74.2
TBW mBIA Handrail (kg)	27.1 ± 5.4	23.3–30.5	32.4 ± 7.4	26.2–37.9	29.9 ± 7.0	15.4–49.5
ASMM mBIA Handrail (kg)	14.87 ± 3.9	11.8–16.6	19 ± 5.5	14.6–22.4	16.6 ± 5.0	6.7–30.8
PA mBIA Handrail	4.8 ± 0.5	4.4–5.1	5.2 ± 0.7	4.7–5.6	5.0 ± 0.6	3.7–6.9
R50T mBIA Handrail (Ω)	26.4 ± 3.3	19.3–36.8	24.5 ± 3.5	17.5–35.2	25.4 ± 3.5	17.5–36.8
Xc50T mBIA Handrail (Ω)	−2.7 ± 0.4	−3.7–−1.8	−3.0 ± 0.4	−4.2–−2.0	−2.9 ± 0.4	−4.1–−1.8
BIA supine 8 electrodes	*n* = 220		*n* = 224		*n* = 444	
FM mBIA supine 8E (kg)	16 ± 9.1	8.5–22.3	14.7 ± 9	6.7–21.4	15.3 ± 9.1	1.8–41.9
FFM mBIA supine 8E (kg)	32.3 ± 10.3	23.0–40.1	39 ± 14	26.0–49.7	35.7 ± 12.7	13.3–71.8
TBW mBIA supine 8E	23.2 ± 7.1	16.9–28.8	28.4 ± 9.6	19.4–36.0	25.8 ± 8.8	9.6–48.8
ASMM mBIA supine 8E (kg)	12.2 ± 4.8	7.7–15.9	16 ± 6.7	9.7–20.6	14.0 ± 6.0	3.8–30.6
PA mBIA supine 8E	5.5 ± 0.5	5.1–5.9	5.9 ± 0.8	5.3–6.4	5.7 ± 0.7	4.3–8.3
R50T mBIA supine 8E (Ω)	32.1 ± 4.2	23.3–45.8	29.6 ± 4.4	20.5–41.2	30.8 ± 4.5	20.5–45.8
Xc50T mBIA supine 8E (Ω)	−3.9 ± 0.5	−5.6–−1.9	−4 ± 0.4	−5.1–−2.9	−3.9 ± 0.5	−5.6–−1.9
BIA supine 4 electrodes RS	*n* = 225		*n*= 224		*n*= 449	
FM mBIA supine 4E RS (kg)	15.5 ± 9.4	7.9–22.1	14.6 ± 9.1	6.9–21.6	15.1 ± 9.2	1.5–43.1
FFM mBIA supine 4E RS (kg)	32.1 ± 10.2	23.2–39.8	39.1 ± 13.7	26.7–49.8	35.6 ± 12.6	13.0–73.1
TBW mBIA supine 4E RS	23.1 ± 6.9	17.1–28.7	27.9 ± 9.1	19.7–35.5	25.5 ± 8.4	8.8–48.3
PA mBIA supine 4E RS	5.5 ± 0.6	5.1–6.0	5.9 ± 0.8	5.3–6.5	5.7 ± 0.7	4.1–8.4
R50RS mBIA supine 4E RS (Ω)	735 ± 101.4	526–1110.5	656.8 ± 103.7	431.0–935.3	696. ± 109.7	431.0–1110.5
Xc50RS mBIA supine 4E RS (Ω)	−70.6 ± 8	−94.2–−53.0	−67.1 ± 7.1	−86.5–−50.2	−68.9 ± 7.8	−94.2–−50.2

Data shown as mean and standard deviation (SD). Abbreviations: BMI, body mass index; DXA, dual X-ray absorptiometry; BMC, bone mineral content; FM, fat mass; FFM, fat-free mass; ALM, appendicular lean mass; mBIA, multifrequency bioelectrical impedance analysis; TBW, total body water; ASMM, appendicular skeletal muscle mass; R, resistance; Xc, reactance; PA:phase angle; RS: right side.

**Table 3 nutrients-14-00965-t003:** Correlation, concordance, agreement, and bias assessment of total FM between BIA methods with respect to DXA.

Method	Correlation r (95% CI)	Lin’s Concordance Coefficient (95% CI)	Bland-Altman Dif Mean ± SD	LOA	Regression Equation	*p* Value
Total sample						
mBIA Handle	0.99 (0.99–0.99)	0.93 (0.92–0.94)	−3.2 ± 1.4	(−6.0 to −0.4)	Y = −0.06899X − 2.071	<0.001
mBIA Handrail	0.99 (0.99–0.99)	0.98 (0.97–0.98)	−3.6 ± 1.5	(−6.5 to −0.7)	Y = −0.07939X − 2.055	<0.001
mBIA supine 8e	0.99 (0.99–0.99)	0.95 (0.94–0.96)	−2.6 ± 1.1	(−5.0 to −0.7)	Y = −0.03977X − 2.180	<0.001
mBIA supine 4e	0.99 (0.99–0.99)	0.94 (0.94–0.95)	−3.0 ± 1.3	(−5.4 to −0.5)	Y = −0.02697X − 2.507	<0.001
NW						
mBIA Handle	0.98 (0.97–0.99)	0.83 (0.79–0.86)	−2.9 ± 1.0	(−4.8 to −0.9)	Y = −0.02303X − 2.660	0.17
mBIA Handrail	0.97 (0.96–0.98)	0.95 (0.93–0.97)	−3.0 ± 1.2	(−5.3 to −0.6)	Y = −0.03366X − 2.591	0.2
mBIA supine 8e	0.98 (0.98–0.99)	0.87 (0.83–0.89)	−2.7 ± 0.9	(−4.5 to −0.9)	Y = −0.04740X − 2.270	0.002
mBIA supine 4e	0.98 (0.97–0.98)	0.84 (0.81–0.87)	−2.8 ± 1.1	(−4.9 to −0.6)	Y = −0.06141X − 2.189	<0.001
OW						
mBIA Handle	0.99 (0.98–0.99)	0.86 (0.83–0.89)	−3.2 ± 1.1	(−5.4 to 1.1)	Y = −0.05292X − 2.402	<0.001
mBIA Handrail	0.98 (0.97–0.98)	0.93 (0.91–0.94)	−3.6 ± 1.1	(−5.9 to −1.4)	Y = −0.04778X − 2.753	0.03
mBIA supine 8e	0.99 (0.98–0.99)	0.9 (0.87–0.92)	−3.0 ± 1.1	(−5.0 to −0.9)	Y = −0.05542X − 2.060	<0.001
mBIA supine 4e	0.98 (0.98–0.99)	0.89 (0.86–0.91)	−3.0 ± 1.3	(−5.5 to −0.6)	Y = −0.03299X − 2.511	0.04
OB						
mBIA Handle	0.98 (0.98–0.99)	0.89 (0.86–0.91)	−3.5 ± 1.9	(−7.2 to 0.1)	Y = −0.1467X + 0.07549	<0.001
mBIA Handrail	0.98 (0.97–0.98)	0.91 (0.88–0.92)	−4.1 ± 1.8	(−7.5 to −0.6)	Y = −0.1559X + 0.1657	<0.001
mBIA supine 8e	0.99 (0.99–0.99)	0.93 (0.91–0.95)	−2.9 ± 1.3	(−5.5 to −0.3)	Y = −0.07974X − 0.9045	<0.001
mBIA supine 4e	0.99 (0.98–0.99)	0.93 (0.91–0.94)	−3.1 ± 1.4	(−5.8 to −0.4)	Y = −0.02301X − 2.511	0.08

CI, confidence interval; SD standard deviation; NW, normal weight; OW, overweight; OB, obesity; BA, Bland Altman; LOA, limits of agreement.

**Table 4 nutrients-14-00965-t004:** Correlation, concordance, agreement, and bias assessment of total FFM between BIA methods with respect to DXA.

Method	Correlation r (95% CI)	Lin’s Concordance Coefficient (95% CI)	Bland-Altman Dif Mean ± SD	LOA	Regression Equation	*p* Value
Total sample						
mBIA Handle	0.99 (0.99–1.0)	0.96 (0.96–0.97)	3.3 ± 1.5	(0.4 to 6.2)	Y = 0.05619X + 1.399	<0.001
mBIA Handrail	0.99 (0.99–0.99)	0.75 (0.71–0.78)	3.7 ± 1.6	(0.6 to 6.8)	Y = 0.05275X + 1.598	<0.001
mBIA supine 8e	1.00 (0.99–1.0)	0.97 (0.96–0.97)	3.0 ± 1.2	(0.6 to 5.3)	Y = 0.02105X + 2.230	<0.001
mBIA supine 4e	0.99 (0.99–1.0)	0.97 (0.96–0.97)	3.1 ± 1.4	(0.4 to 5.7)	Y = −0.001829X + 3.119	0.72
NW						
mBIA Handle	1.00 (0.99–1.0)	0.97 (0.96–0.98)	2.8 ± 1.0	(0.8 to 4.9)	Y = 0.002063X + 2.767	0.79
mBIA Handrail	0.99 (0.98–0.99)	0.65 (0.58–0.72)	2.9 ± 1.3	(0.5 to 5.4)	Y = −0.03812X + 4.298	0.01
mBIA supine 8e	1.00 (0.99–1.0)	0.96 (0.95–0.97)	2.7 ± 1.0	(0.7 to 4.6)	Y = −0.01039X + 2.979	0.16
mBIA supine 4e	0.99 (0.99–1.0)	0.97 (0.96–0.97)	2.7 ± 1.2	(0.4 to 5.0)	Y = −0.01672X + 3.204	0.06
OW						
mBIA Handle	1.00 (0.99–1.0)	0.96 (0.95–0.97)	3.3 ± 1.2	(1.0 to 5.6)	Y = 0.02910X + 2.315	<0.001
mBIA Handrail	0.99 (0.99–0.99)	0.73 (0.67–0.78)	3.7 ± 1.2	(1.3 to 6.1)	Y = 0.01966X + 2.949	0.12
mBIA supine 8e	1.00 (0.99–1.0)	0.96 (0.95–0.97)	3.0 ± 1.2	(0.8 to 5.3)	Y = 0.009462X + 2.703	0.2
mBIA supine 4e	0.99 (0.99–1.0)	0.96 (0.95–0.97)	3.1 ± 1.4	(0.5 to 5.8)	Y = −0.01198X + 3.521	0.2
OB						
mBIA Handle	1.00 (0.99–1.0)	0.96 (0.94–0.97)	3.8 ± 1.9	(0.1 to 7.5)	Y = −0.1467X + 0.07549	<0.001
mBIA Handrail	0.99 (0.99–0.99)	0.79 (0.74–0.83)	4.4 ± 1.8	(0.8 to 7.9)	Y = −0.1559X + 0.1657	<0.001
mBIA supine 8e	1.00 (0.99–1.0)	0.97 (0.96–0.98)	3.2 ± 1.4	(0.4 to 5.9)	Y = −0.07974X − 0.9045	<0.001
mBIA supine 4e	0.99 (0.99–1.0)	0.93 (0.91–0.94)	3.4 ± 1.4	(0.5 to 6.2)	Y = −0.02301X − 2.511	0.08

CI, confidence interval; SD standard deviation; NW, normal weight; OW, overweight; OB, obesity; LOA, limits of agreement.

**Table 5 nutrients-14-00965-t005:** Correlation, concordance, agreement, and bias assessment of ALM between BIA methods with respect to DXA.

Method	Correlation r (95% CI)	Lin’s Concordance Coefficient (95% CI)	Bland-Altman Dif Mean ± SD	LOA	Regression Equation	*p* Value
Total sample						
mBIA Handle	0.99 (0.994–0.995)	0.99 (0.99–1.0)	0.61 ± 0.65	(−0.66–1.87)	Y = 0.03894X + 0.1193	<0.001
mBIA Handrail	0.99 (0.988–0.99)	0.87 (0.85–0.89)	0.49 ± 0.72	(0.45 to 0.71)	Y = 0.01073X + 0.2830	0.009
mBIA supine 8e	0.99 (0.99–0.995)	0.99 (0.99–0.99)	0.02 ± 0.59	(0.17 to 0.63)	Y = 0.006326X + 0.08503	0.009
NW						
mBIA Handle	0.99 (0.992–1.00)	0.99 (0.985–0.992)	0.64 ± 0.61	(0.985–0.992)	Y = 0.03909X + 0.1751	<0.001
mBIA Handrail	0.99 (0.98–0.99)	0.77 (0.682–0.830)	0.49 ± 0.72	(−0.93 to 1.91)	Y = −0.002198X + 0.5207	0.92
mBIA supine 8e	0.99 (0.99–0.995)	0.99 (0.987–0.993)	0.02 ± 0.59	(−0.96 to 1.36)	Y = 0.008425X + 0.1044	0.38
OW						
mBIA Handle	0.99 (0.99–0.995)	0.99 (0.985–0.992)	0.64 ± 0.7	(−0.74 to 2.01)	Y = 0.04478X + 0.004772	<0.001
mBIA Handrail	0.99 (0.987–0.995)	0.83 (0.772–0.878)	0.40 ± 0.6	(−0.78 to 1.58)	Y = 0.01184X + 0.1981	0.47
mBIA supine 8e	0.99 (0.99–0.995)	0.99 (0.990–0.995)	0.14 ± 0.7	(−1.21 to 1.48)	Y = 0.01138X − 0.02175	0.26
OB						
mBIA Handle	1.00 (0.993–0.996)	0.99 (0.986–0.993)	0.75 ± 0.70	(−0.63 to 2.12)	Y = 0.04722X − 0.03256	<0.001
mBIA Handrail	0.99 (0.989–0.996)	0.88 (0.835–0.908)	0.57 ± 0.69	(−0.77 to 1.92)	Y = 0.01259X + 0.3321	0.36
mBIA supine 8e	0.99 (0.991–0.995)	0.99 (0.991–0.995)	0.08 ± 0.75	(−1.39 to 1.56)	Y = 0.01588X − 0.1738	0.09

CI, confidence interval; SD standard deviation; NW, normal weight; OW, overweight; OB, obesity; LO limits of agreement.

**Table 6 nutrients-14-00965-t006:** Correlation, concordance, agreement, and bias assessment of tFM between BIA methods with respect to DXA.

Method	Correlation r (95%CI)	Lin’s Concordance Coefficient (95%CI)	Bland-Altman Dif Mean ± SD	LOA	Regression Equation	*p* Value
Total sample						
mBIA Handle	0.99 (0.99–0.99)	0.99 (0.99–0.99)	−0.18 ± 0.86	(−1.9 to 1.5)	Y = −0.01036X − 0.08980	*p* value = 0.18
mBIA Handrail	0.98 (0.98–0.99)	0.94 (0.93–0.95)	−0.23 ± 0.97	(−2.1 to 1.7)	Y = −0.03417X + 0.1294	*p* value = 0.002
mBIA supine 8e	0.99 (0.99–0.99)	0.99 (0.99–0.99)	−0.09 ± 0.8	(−1.7 to 1.53)	Y = −0.02612X + 0.1470	*p* value <0.001
mBIA supine 4e	0.99 (0.98–0.99)	0.99 (0.98–0.99)	−0.08 ± 0.96	(−2.0 to 1.8)	Y = −0.007186X − 0.01768	*p* value = 0.39
NW						
mBIA Handle	0.97 (0.96–0.98)	0.97 (0.96–0.98)	−0.13 ± 0.67	(−1.5 to 1.2)	Y = −0.04764X + 0.07921	*p* value = 0.02
mBIA Handrail	0.96 (0.94–0.98)	0.91 (0.87–0.93)	−0.07 ± 0.8	(−1.7 to 1.5)	Y = −0.01808X + 0.02862	*p* value = 0.65
mBIA supine 8e	0.98 (0.97–0.99)	0.98 (0.97–0.98)	−0.01 ± 0.59	(−1.16 to 1.13)	Y = −0.03642X + 0.1521	*p* value = 0.04
mBIA supine 4e	0.96 (0.95–0.97)	0.97 (0.95–0.98)	0.03 ± 0.78	(−1.5 to 1.56)	Y = −0.07618X + 0.3707	*p* value < 0.001
OW						
mBIA Handle	0.97 (0.96–0.98)	0.97 (0.96–0.98)	−0.23 ± 0.83	(−1.86 to 1.4)	Y = −0.005690X − 0.1789	*p* value = 0.78
mBIA Handrail	0.95 (0.92–0.97)	0.87 (0.83–0.90)	−0.12 ± 0.93	(−1.95 to 1.71)	Y = 0.03249X − 0.4467	*p* value = 0.42
mBIA supine 8e	0.98 (0.97–0.98)	0.98 (0.97–0.98)	−0.08 ± 0.8	(−1.64 to 1.49)	Y = −0.03175X + 0.1897	*p* value = 0.08
mBIA supine 4e	0.97 (0.96–0.98)	0.97 (0.96–0.98)	−0.13 ± 0.96	(−2.0 to 1.74)	Y = −0.01300X − 0.01975	*p* value = 0.54
OB						
mBIA Handle	0.98 (0.97–0.98)	0.98 (0.97–0.98)	−0.14 ± 1.04	(−2.18 to 1.9)	Y = −0.009156X − 0.009721	*p* value = 0.63
mBIA Handrail	0.96 (0.94–0.98)	0.89 (0.85–0.92)	−0.38 ± 1.1	(−2.58 to 1.82)	Y = −0.01641X − 0.1278	*p* value = 0.6
mBIA supine 8e	0.98 (0.97–0.98)	0.98 (0.97–0.99)	−0.18 ± 1.03	(−2.19 to 1.83)	Y = −0.03294X + 0.2764	*p* value = 0.06
mBIA supine 4e	0.98 (0.97–0.98)	0.97 (0.97–0.98)	−0.12 ± 1.13	(−2.33 to 2.09)	Y = 0.02225X − 0.4251	*p* value = 0.23

CI, confidence interval; SD standard deviation; NW, normal weight; OW, overweight subsample; OB, obese subsample; LOA, limits of agreement.

## Data Availability

Not applicable.

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
