# Peer review of "Body Composition Assessment in Mexican Children and Adolescents. Part 2: Cross-Validation of Three Bio-Electrical Impedance Methods against Dual X-ray Absorptiometry for Total-Body and Regional Body Composition"

_nutrients, 2022, doi:10.3390/nu14050965_

Round 1

Reviewer 1 Report

Dear Authors,

I have had the opportunity to read your manuscript entitled “Body composition assessment in Mexican children and adolescents. Part 2: Cross-validation of three bio-electrical impedance methods against dual X-ray absorptiometry for whole-body and regional body composition

    This work offers an exhaustive comparison among three different BIA methods for estimating body composition in children an adolescents. The study shows remarkable strengths:

  • The evenly distribution of healthy weight, overweight and obese subjects within the sample as well as the distribution according to the Tanner puberal stages, confers a huge consistency to each one of the estimations performed.
  • The methodology has been explained in a detailed and comprehensive way, which makes understandable the analysis.
  • Tables and figures are clear and well presented.
  • The discussion summarizes the current evidence available and compares their own findings with the preexisting ones. The limitations of the study have been pointed out properly.

I just have a couple of little comments:

  • Line 33: the expression “several health outcomes” is very uncertain, imprecise. I suggest that probably some comments should be added, to define the usefulness of the body composition assessment.
  • I would include in the abstract that this validation cannot be extrapolated to other BIA devices. It is an important statement which should be highlighted.

Author Response

We, the authors, want to express our gratitude to the Editorial Board and the experts that reviewed our manuscript for their contributions. We have found your recommendations and commentaries of great value and have made appropriate changes. We feel confident that the manuscript has been strengthened by this.

Please find our answers in the following lines.

Comments:

  1. Line 33: the expression “several health outcomes” is very uncertain, imprecise. I suggest that probably some comments should be added, to define the usefulness of the body composition assessment.

Added, lines 34-35

2. I would include in the abstract that this validation cannot be extrapolated to other BIA devices. It is an important statement which should be highlighted.

Added, line 23

Reviewer 2 Report

I didn't fully understand the methods used in this study, which could be due to my own limitations with the content. Primarily, I did not understand whether the data presented in Tables 2-5 were based on the reference or the validation samples. Was the only data presented on the reference sample in the Supplement? And was methods section 2.4.1 undertaken only on the reference sample? And then equations were developed based on that data and applied to the main sample to create the various estimates shown in Tables 2-5? Was the 2nd paragraph of the Results (lines 144-149) referring only to the reference sample? Assuming I've interpreted the various sections correctly, I don't have any other real criticisms of the study, but  I also clearly lack the expertise to fully review their methods, nor do I feel qualified to judge the importance of the findings. 

Author Response

We, the authors, want to express our gratitude to the Editorial Board and the experts that reviewed our manuscript for their contributions. We have found your recommendations and commentaries of great value and have made appropriate changes. We feel confident that the manuscript has been strengthened by this.

Please find our answers in the following lines.

Comments:

I didn't fully understand the methods used in this study, which could be due to my own limitations with the content. Primarily,

1. I did not understand whether the data presented in Tables 2-5 were based on the reference or the validation samples. Was the only data presented on the reference sample in the Supplement?

We appreciate the very relevant observation and we have corrected accordingly. Specifically, we added relevant descriptors within the text and tables’ titles with corresponding terms “reference sample” or “validation sample”. We also moved the supplementary table 1 to the body of the article and now it is Table 1.

2. And was methods section 2.4.1 undertaken only on the reference sample?

Yes, this is correct. We have worked to improve the clarity of such description.

3. And then equations were developed based on that data and applied to the main sample to create the various estimates shown in Tables 2-5?

Yes, this is correct. Tables 2-5 (before changes, now 3-6), show the relevant data for the validation sample (N=450).

4. Was the 2nd paragraph of the Results (lines 144-149) referring only to the reference sample?

Yes, we added, line 152